# Immunohistochemical Markers and TILs Evaluation for Endometrial Carcinoma

**DOI:** 10.3390/jcm11195678

**Published:** 2022-09-26

**Authors:** Valentina Elisabetta Bounous, Annamaria Ferrero, Paola Campisi, Luca Fuso, Jeremy Oscar Smith Pezua Sanjinez, Sabrina Manassero, Giovanni De Rosa, Nicoletta Biglia

**Affiliations:** 1Gynecology and Obstetrics Unit, Umberto I Hospital, Department of Surgical Sciences, University of Turin, Largo Turati 62, 10128 Turin, Italy; 2Department of Pathology, Umberto I Hospital, Largo Turati 62, 10128 Turin, Italy

**Keywords:** endometrial cancer, immunohistochemistry, estrogen receptors, mismatch repair, disease-free survival, outcome

## Abstract

Objective: The molecular classification for endometrial cancer (EC) introduced by The Cancer Genome Atlas Research Network (TCGA) and the Proactive Molecular Risk Classifier for Endometrial Cancer (ProMisE) proved the existence of four molecular prognostic subtypes; however, both classifications require costly technology. We suggest a prognostic model for EC based on immunohistochemistry (IHC) and tumor-infiltrating lymphocytes (TILs). Study design: One hundred patients were included. We retrospectively investigated IHC prognostic parameters: mismatch repair (MMR)-deficient tumors, p53 mutation status, progesterone receptors (PgRs), and estrogen receptors (ERs). We further evaluated TILs. These parameters were related to the clinical and morphological features and to the outcome. Results: We classified tumors into three groups (IHC analysis): MMR-deficient, p53-mutated, p53 wild-type. MMR-deficient tumors had a good prognosis, p53 wild-type tumors an intermediate one, and p53-mutated tumors had the poorest outcomes. Disease-free (DFS) and overall survival (OS) were significantly better among PgR+ tumors (respectively *p* = 0.011 and *p* = 0.001) and PgR expression is an independent prognostic factor for a better DFS frommultivariate analysis (OR = 0.3; CI: 0.1–0.9; *p* = 0.03).No significant correlation was observed between DFS and TILs. However, among MMR-deficient tumors, the mean value of TILs was higher than among the other tumors(111 versus 71, *p* = 0.01) Conclusions: The prognostic model based on IHC markers could potentially be a valid and applicable alternative to the TCGA one. The PgR determination could represent an additional prognostic factor for EC.

## 1. Introduction

Although endometrial cancer (EC) has a low recurrence rate and mostly good outcomes [1,2,3,4], identification of high-risk patients is an urgent research priority [5] to plan adjuvant treatment. It is current knowledge that the dualistic model [6,7,8] is no longer able to classify EC in an adequate way [9].

In 2013, a genomic, proteomic and transcriptomic characterization of EC introduced by The Cancer Genome Atlas Research Network (TCGA) proved the existence of four molecular prognostic subtypes [10], adding new prognostic and predictive information, giving us the ability to identify high-risk patients [1]. However, the sequencing-based technologies used in the TCGA study require complex and expensive techniques that cannot be afforded by all hospital departments.

In 2018, the validation of the Proactive Molecular Risk Classifier for Endometrial Cancer (ProMisE) identified four simplified molecular prognostic subgroups [11], but even this classification requires costly molecular-based technologies.

Hormonal receptors for estrogen (ER) and progesterone (PgR), mutations of p53, mismatch repair (MMR) deficiency, and tumor-infiltrating lymphocytes (TILs), are considered promising markers for EC prognosis [1,12,13,14]. The most recent European guidelines on EC management define MMR status testing as relevant and states that it should be performed in all ECs, irrespective of histologic subtype and patient age. Since it has a prognostic role, it identifies patients at high risk for Lynch syndrome and it can be a predictor of potential utility of immune checkpoint inhibitor therapy [1].

An association between TIL expression and both POLE [15,16] and MMR-deficient tumors has been described in the literature.

In this retrospective study, we investigated TIL expression and a panel of IHC markers (MMR status, ER, PgR, p53) in order to evaluate their prognostic role in terms of disease-free survival (DFS) and overall survival (OS) and assess their relationship with histopathological prognostic factors, such as grading, histotype, FIGO stage at diagnosis, myometrial invasion, and lymphovascular space invasion (LVSI).The aim of our study is to validate in a clinical setting the new prognostic classification for EC with IHC techniques to guide adjuvant treatment choice in an applicable and less expensive way than the one described by TCGA.

## 2. Materials and Methods

### 2.1. Study Population

A retrospective cohort of 100 patients with primary EC surgically treated in our Department between January 2009 and December 2017 was analyzed.

Exclusion criteria included hyperplasia, other cancers metastatic to the uterus, no definitive surgery performed, and lack of all requested data.

All patients signed a written informed consent stating their clinical data and biological material could be used for research purposes. No Ethics Committee approval is required at our institution for retrospective studies on individual patient data collected in anonymized datasets.

### 2.2. IHC Analysis

Formalin-fixed paraffin-embedded tissue was collected from the archives of the pathology division of our hospital. Hematoxylin–eosin archival slides were revised by an expert pathologist and the diagnosis was confirmed in all cases. The most representative slide of each sample was selected for IHC in cases with available material. Specifically, three-micron-thick serial paraffin sections of each case were processed by IHC using an automated immunostainer (Ventana BenchMark Ultra AutoStainer, Ventana Medical Systems, Tucson, AZ, USA) with antibodies against p53 (clone DO-7, catalogue number 7800-2912,Ventana Medical Systems, Tucson, AZ, USA), ER (clone SP1, catalogue number 790-4324, Ventana Medical Systems, Tucson, AZ, USA), PgR (clone 1E2, catalogue number 790-2223, Ventana Medical Systems, Tucson, AZ, USA), and the MMR status, evaluating the protein expression of MLH1 (clone M1, catalogue number 760-5091, Ventana Medical Systems, Tucson, AZ, USA), PMS2 (clone EPR3947, catalogue number 760-5094, Ventana Medical Systems, Tucson, AZ, USA), MSH2 (clone G219-1129, catalogue number 760-5093, Ventana Medical Systems, Tucson, AZ, USA), MSH6 (clone 44, catalogue number 760-5092, Ventana Medical Systems, Tucson, AZ, USA). Appropriate positive controls were included for each IHC run.

#### 2.2.1. MMR Status

We investigated the MMR status by IHC evaluating the presence or absence of proteins involved in the mismatch repair pathway: MLH1, PMS2, MSH2, and MSH6. Considering MLH1 and PMS2 as heterodimers with MLH1 dominance, MLH1 loss confirms the absence of PMS2; nevertheless, PMS2 loss does not determine MLH1 loss. Similarly, MSH2 and MSH6 are partners with MSH2 dominance. MMR deficiency is defined by lack of expression of at least one of these proteins, as previously reported [17].

MMR-IHC is also the recommended test by The International Society of Gynecological Pathology (ISGyP) guidelines to assess MMR status, since it is widely available and cost-effective [18]. In the literature, it is described that IHC evaluation of MMR can be assessed also considering the expression of only MSH2 and PMS2 [19] but in our series, a complete evaluation of MLH1, PMS2, MSH2, and MSH6 was performed.

In Figure 1 and Figure 2, an example of a tumor with MMR deficiency and one without it are shown, respectively.

#### 2.2.2. Hormone Receptors

ER and PgR expression by IHC was reported as negative or positive (≥10%) [20].

#### 2.2.3. p53

p53 nuclear expression by IHC has been considered wild-type (wt) (Figure 3) when focally expressed with variable intensity or mutated when highly expressed in nearly all tumor cells with high intensity, or when totally absent [17] (Figure 4).

It has been demonstrated in the literature that IHC for p53 is a highly accurate surrogate of TP53 sequencing [21].

### 2.3. TILs

We quantitatively evaluated TILs for 10 HPF present in the stromal compartment within the border of the invasive tumor in hematoxylin-and-eosin-stained tumor sections. Lymphocytes and plasma cells were counted while granulocytes were excluded. A consensus recommendation for TIL evaluation is available at the moment only for breast cancer, and we employed it [22].

However, for breast cancer, no formal recommendation for a clinically relevant TIL threshold has existed until now. Due to the lack of a standardized cut-off, we preferred using a continuous parameter, classifying patients with high TIL levels using as a threshold the median value obtained from our results.

### 2.4. Statistical Analysis

All statistical analyses were performed using Windows SPSS software version number 23 created by the International Business Machines Corporation (IBM) in Armonk, NY, USA.

Primary endpoints were DFS and secondary outcome OS. We used the chi-square test to verify the existence of a causal relationship between categorical variables. The Student’s *t*-test and variance analysis were applied considering categorical and continuous variables. Pearson’s correlation was performed relating continuous variables. Starting at the date of the diagnosis, survival was estimated using the Kaplan–Meier method and compared with the log-rank test.

Cox proportional hazard regression was for the multivariate analysis, including univariate statistically significant covariables. Relative risk for death or recurrence was defined through the odds ratio (OR) with a 95% confidence interval (CI 95%). Significance was considered at 5%.

## 3. Results

### 3.1. Clinical and Pathological Features

Clinical and pathological features of the cohort are shown in Table 1.

Mean age at diagnosis was 66 years. Most cases were diagnosed during postmenopausal age, while 7% of patients were younger than 50 years. In 14% of patients, a family history of pelvic or gastrointestinal tumors was reported. The more frequent histotype was endometrioid EC (84% of cases). Most ECs were diagnosed at stage I (67%). All patients underwent surgery (at that time, 70% used a laparotomic approach and 30% a laparoscopic approach, while nowadays we perform 100% laparoscopic as a first approach to EC at our institution). A lymphadenectomy was required and performed in 45% of cases. LVSI was present in one third of cases. Adjuvant treatment was performed according to the European Guidelines for EC treatment currently available during the study period. Half of the patients received adjuvant therapy (chemotherapy, radiotherapy, brachytherapy); one patient also received neoadjuvant chemotherapy.

### 3.2. IHC Features

We identified 22 (22%) MMR-deficient tumors, 22 cases (22%) with p53 mutations, and 56 (56%) with p53 wt. After the IHC analysis, patients were stratified into three subgroups: MMR deficiency, p53 mutation, and p53 wt.

#### 3.2.1. MMR Status

Among MMR-deficienttumors (*n* = 22), the most frequent protein observed to be lost was MLH1 (15/22, 68% of MSI), followed by MSH2 (3/22, 14%), PMS2 (3/22, 14%), and MSH6 (1/22, 4%). Mean age at diagnosis was 65 years (SD ± 10) in both MMR-deficient tumors and in MMR-normal tumors.

No significant differences were observed between MMR status and histopathological features.

#### 3.2.2. p53 Status

Patients with p53-mutated tumors were diagnosed at a later age: 70 years, versus 62 years for patients with p53 wt tumors (*p* = 0.006). Fifty-eight percent of p53-mutated tumors were endometrioid; however, compared to the p53 wt group, a significantly high prevalence of the non-endometrioid histotype was observed (41.7% vs. 10.5%, *p* = 0.0001). Most p53-mutated tumors were high-grade (66.7%) (*p* = 0.004). Approximately 41.7% of p53-mutated tumors were more frequently diagnosed at an advanced stage, while p53 wt neoplasms were diagnosed earlier (*p* =0.023) (Table 2).

#### 3.2.3. Hormone Receptors

A total of 84% of tumors were ER-positive and 85% were PgR-positive. The mean age at diagnosis was 73 years in ER- and PgR-negative patients versus 63 years in ER- and PgR-positive patients (*p* = 0.05).

##### ESTROGEN-RECEPTORS (ER)

A total of 84% of tumors were ER-positive. ER-negative tumors were more often non-endometrioid (*p* = 0.003), high-grade (*p* = 0.011), at an advanced FIGO stage (*p* = 0.013), and p53-mutated (*p* = 0.019)(Table 3).

##### PROGESTERONE RECEPTOR PgR

PgR negative tumors were more often non-endometrioid (*p* = 0.029) and high-grade (*p* = 0.003) tumors (Table 4).

#### 3.2.4. TILs

The TIL mean level was 82/10 HPF (SD ± 59), and the median value was 60/10 HPF (range 10–300).

High levels of TILs, defined as TILs over the median value, did not show a significant statistical correlation with histology, tumor grade, or FIGO stage. However, among MMR-deficient tumors, the mean value of TILs was significantly higher than the other tumors (111 versus 71, *p* = 0.01).

### 3.3. Outcome

Median follow-up time was 42 months (mean time 54 months). At the last follow-up, 77 (77%) of the patients were in complete remission, 16 (16%) died because of the disease, and 1for other reasons; 2 (2%) patients were undergoing treatment for recurrence, and 4 were in complete remission after recurrence; 19 (19%) had total recurrences, 10 (52.6%) distant recurrences, and 9 (47.4%) locoregional recurrences.

#### 3.3.1. DFS

As reported in Figure 5, the three IHC subgroups had different prognoses: the MMR-deficient group showed the best outcome while the p53-mutated one had the poorest; p53 wt tumors had an intermediate prognosis (*p* = 0.05).

A better DFS was observed in the early compared to advanced stages (*p* = 0.00001), in endometrioid compared to non-endometrioid histology (*p* = 0.00001), and in PgR+ versus PgR− tumors (*p* = 0.011) (Figure 6). On the contrary, high-grade tumors (*p* = 0.00001) were significantly related to a shorter DFS.

No significant correlation was observed between DFS and TILs.

Univariate survival analysis confirms that high grade (G3 vs. G1-2 OR = 10, 95% CI: 2.8–35.4; *p* = 0.00001), advanced FIGO stage (advanced vs. early OR = 2.9, 95% CI: 1.7–4.7; *p* = 0.0001), and the presence of LVSI (OR = 4.2, 95% CI: 2.6–4.5; *p* = 0.006) were correlated with worse DFS. Particularly, p53 mutation determined higher recurrence risk (OR = 3.1, 95% CI: 1.1–8.5; *p* = 0.02), while PgR+ and ER+ tumors were correlated with better DFS (OR = 0.2, 95% CI: 0.08–0.5; *p* = 0.003 and 0.3, 95% CI: 0.1–0.9; *p* = 0.04, respectively).

During multivariate analysis, PgR expression was related to a better DFS (OR = 0.3; CI: 0.1–0.9; *p* = 0.03).

#### 3.3.2. OS

OS was significantly better among PgR+ tumors (*p* = 0.001), Figure 7.

OS for patients with p53 wt tumors was significantly better than for those with p53-mutated tumors (*p* = 0.017), as shown in Figure 8, while no significant difference in terms of OS was observed related to MMR status.

High-grade (*p* = 0.000001), advanced-stage (*p* = 0.000001), and non-endometrioid (*p* = 0.000001) tumors and the presence of LVSI (*p* = 0.003) were significantly associated with a shorter OS.

No significant correlation was observed between OS and TIL expression.

## 4. Discussion

EC is the most common pelvic malignancy in the developed world, being the fourth most common cancer in women [23]. Its incidence has increased in the last decades due to the spread of obesity in industrialized countries, representing one of the prevalent risk factors for EC [2]. Most ECs are sporadic, with an estimated 5% occurring in the context of Lynch syndrome, which is an hereditary cancer predisposition caused by mutation in one of the MMR genes, which also confers an increased risk of colon cancer [24]. MMR status has been added to the most recent guidelines on EC management as relevant for its prognostic role, for correctly deciding post-surgical treatment, and to identify high-risk families for Lynch syndrome [1]. In 80% of patients, the disease presents at an early stage, with 5 years overall survival ranging from 74% to 91% [4]. Although EC has a low recurrence rate and mostly good outcomes, identification of high-risk patients is an urgent research priority to plan a correct adjuvant treatment, avoiding overtreatment for low-risk patients (de-escalation), and intensifying management for those at high risk [5,25,26]. EC is a heterogeneous disease. Since the dualistic model for EC proposed by Bokhman in 1983 [27] can no longer precisely discriminate among different EC types, in 2013, a molecular classification was proposed by the TCGA group [10]. The new molecular TGCA classification proved the existence of four molecular subtypes of EC, which related to different outcomes. They examined 373 ECs with complex genome-based techniques, including proteomic and transcriptomic analysis. The study required fresh-frozen tissues and identified four main genomic alterations summarized in four subgroups: POLE ultramutated, MSI hypermutated, Copy Number Low (CNL), and Copy Number High (CNH). They showed that endometrioid tumors mostly belong to the POLE, MSI, and CNL groups. CNL tumors present high ER and PgR expression in the absence of p53 abnormality. In contrast, serous-like and high-grade endometroid tumors are identified in the CNH group, with the poorest outcomes; these tumors are characterized by low ER and PgR expression scores and an abnormal p53 with very aggressive behavior.

A similar EC classifier was published by ProMisE, with three main differences: they used formalin-fixed paraffin-embedded tissues instead of fresh-frozen ones. The identification of the POLE group still required genome-based technologies, but the molecular techniques applied for the other three groups were replaced by IHC analysis. The new four molecular subgroups are similar but not identical to the TCGA ones, demonstrating they are reproducible and applicable both at diagnostic and staging time, although they still require costly and complex methodologies [11].

Our study aimed to apply in a clinical setting the new prognostic classification for EC, employing less expensive technologies than the ones described by TCGA and ProMISE, which are not available in all hospital departments treating EC.

In our study, we evaluated in a series of 100 patients surgically treated for EC MMR status by IHC and other markers (p53 and hormonal receptors by IHC), to evaluate their prognostic role in terms of DFS and OS and to assess their relationship with histopathological prognostic factors. Patients were divided into three subgroups based on IHC results: MMR-deficient tumors (*n* = 22), p53-mutated tumors (*n* = 22), and p53 wt (*n* = 56).

Furthermore we evaluated TIL expression since an association between TILs and both POLE [15,16] and MMR-deficient tumors has been described in the literature [28,29].

In our series, MMR-deficient tumors showed the best DFS compared to the other subtypes with a lower recurrence rate. Furthermore, mean TIL value was significantly higher in this group compared to other tumor groups. This finding is in accordance with the literature in which high TIL expression is correlated to MMR-deficient tumors [30]. Since MMR-deficient tumors are associated with an increased number of mutations (100–1000 fold) and express high levels of neoantigens, an immune microenvironment reaction is observed, with a compensatory upregulation of immune checkpoint proteins (programmed cell death protein 1, PD-1) and their ligands (programmed cell death ligand 1, PD-L1). The combination of this increased mutational load, the presence of TILs, and of PD-1 and PD-L1 expression makes MMR-deficient EC an ideal candidate for immunotherapy [31,32], and for this reason it is important to carry on studies on this topic and establish EC guidelines for TIL evaluation. The lack of a defined cut-off in the literature makes it more difficult to use TILs as prognostic biomarkers at the moment [22]; however, the ability of TILs to explain tumoral antigenic power represents a promising research area.

In our study, p53-mutated tumors had the poorest prognosis, with more than one-third of patients dead because of the disease. Unfavorable tumoral features such as a high grade confirm the negative prognostic impact of p53 mutation on DFS and OS. Tumors with p53 mutated status have aggressive features (non-endometrioid histotype, high grade, hormone negativity, late age of onset, advanced stage at diagnosis), with poor OS, as reported in the literature [33,34].

p53 wt tumors lie in an intermediate position, including patients without MMR deficiency or a p53-mutated status.

On the contrary, high PgR expression is an independent predictive factor of longer DFS during multivariate analysis, being related to favorable EC features (endometrioid histology and low grade),and having a better OS, both in our study and through the literature [35,36].

The main limits of the study are the small sample and the absence of determination of POLE, which requires expensive genomic technologies not available in all hospitals treating EC.

The main strength of the study is indemonstrating that even IHC techniques, which can be capillary diffused and less costly, can reflect different EC subtypes and can be a valid practical tool in the daily life of departments treating EC.

Our findings confirm the hypothesis that there are different prognostic subgroups of EC which could also be determined using IHC techniques.

## Figures and Tables

**Figure 1 jcm-11-05678-f001:**
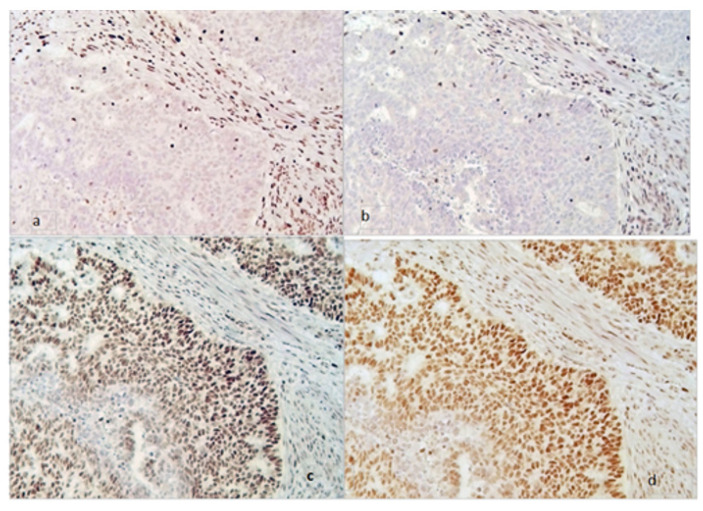
MMR deficiency tumor (magnification 200×). Loss of MSH2 (**a**) and MSH6 (**b**) expression in neoplastic cells with internal positive control of stromal cells (stained brown). Persistence of MLH1 (**c**) and PMS2 (**d**) both in the neoplastic and in the stromal cells (stained brown).

**Figure 2 jcm-11-05678-f002:**
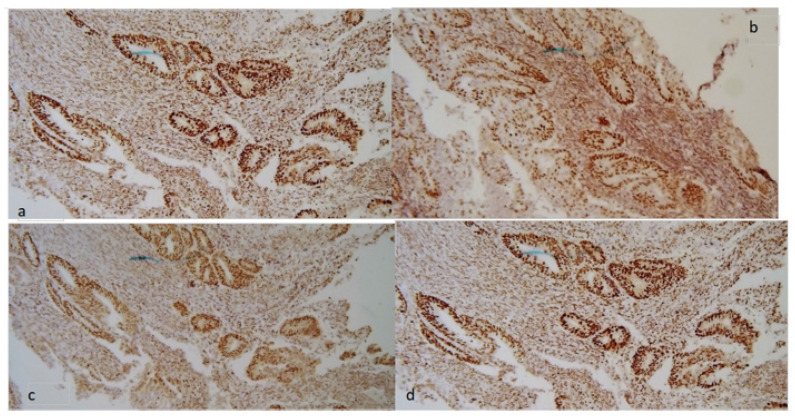
Tumor without MMR deficiency. Normal expression of MMR proteins: MSH2 (**a**), MSH6 (**b**), MLH1 (**c**), and PMS2 (**d**) both in neoplastic cells and in stromal cells (stained brown).

**Figure 3 jcm-11-05678-f003:**
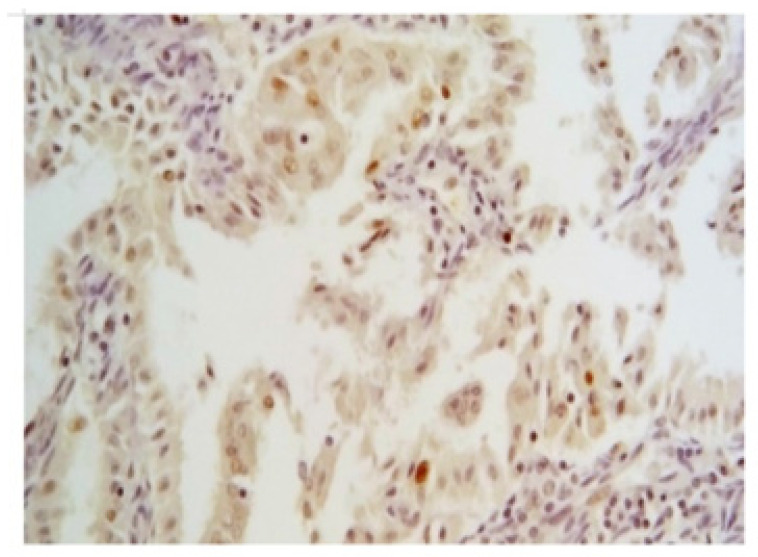
Wild-type pattern of p53 by IHC (magnification 200×). It is characterized by a focal expression with variable but bland intensity in nuclei of neoplastic cells.

**Figure 4 jcm-11-05678-f004:**
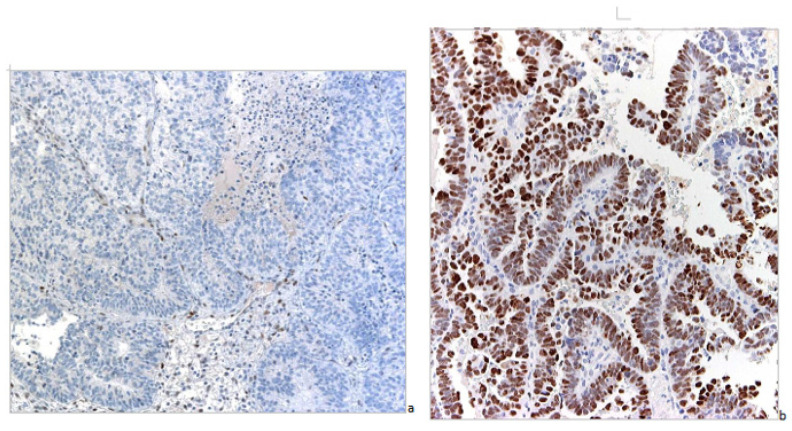
Mutated pattern of p53 by IHC (magnification 100×). (**a**) Neoplastic cells have totally lost expression of the protein. (**b**) Neoplastic cells show intense and diffuse nuclear positivity.

**Figure 5 jcm-11-05678-f005:**
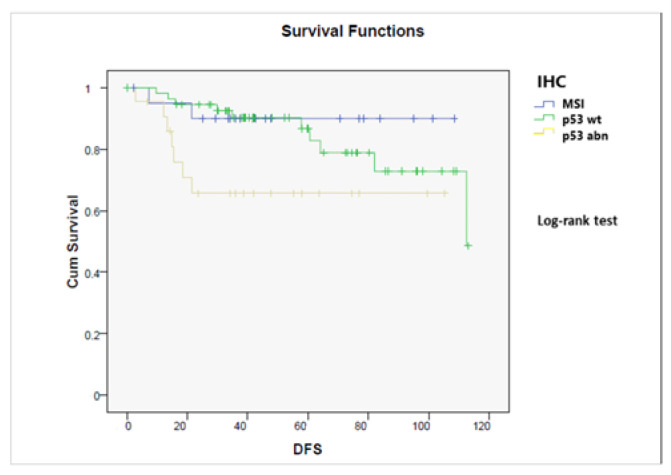
Disease-free survival stratified by immunohistochemical (IHC) status (MMR-deficient, p53-mutated, p53 wild-type) (*p* = 0.05).

**Figure 6 jcm-11-05678-f006:**
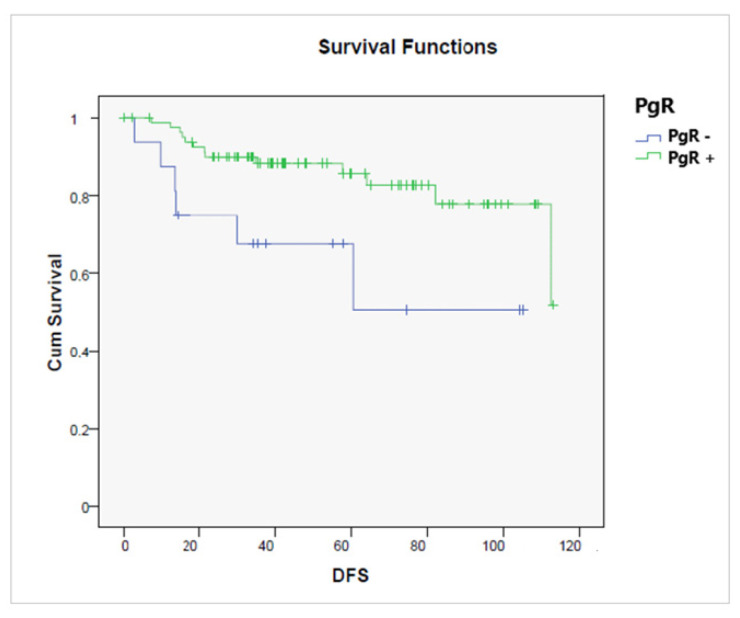
Disease-free survival according to the presence or absence of a progesterone receptor (PgR− vs. PgR+, *p* = 0.011).

**Figure 7 jcm-11-05678-f007:**
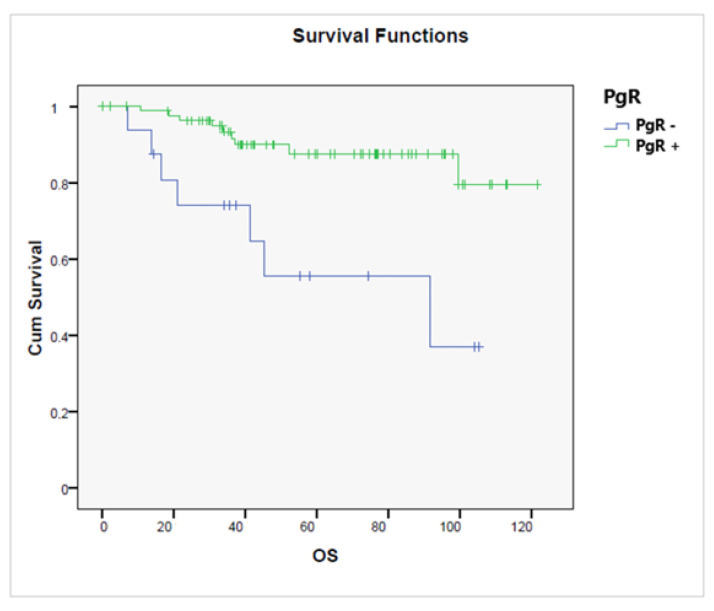
Overall survival according to the presence or absence of a progesterone receptor (PgR+ vs. PgR−, *p* = 0.001).

**Figure 8 jcm-11-05678-f008:**
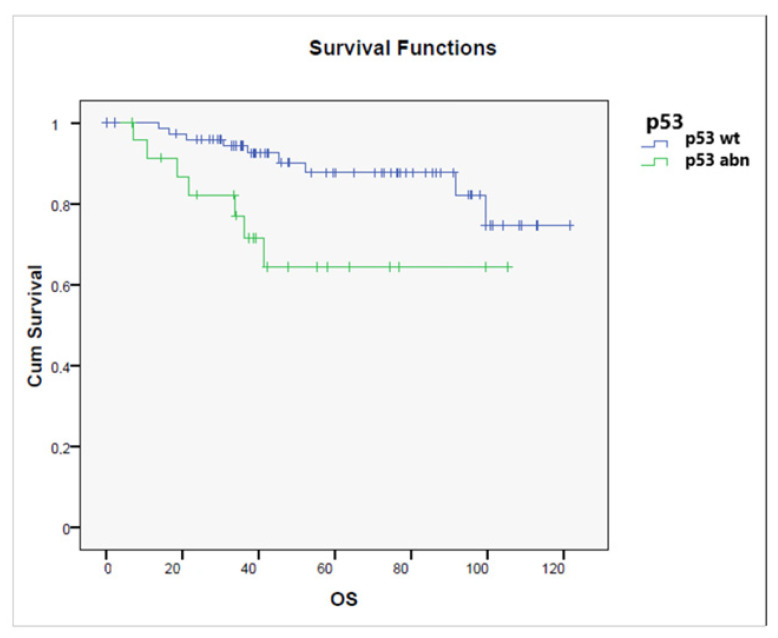
Overall survival according to the status of p53 status (p53 wild-type vs. p53 abnormal) (*p* = 0.017).

**Table 1 jcm-11-05678-t001:** Clinical and pathological features of the study population.

Clinical and Pathological Features		*n* (%) *n* = 100
**AGE**	MEDIAN (RANGE)	65 (36–87)
	MEAN (SD)	66 (SD ± 10)
	DIAGNOSIS BEFORE 50 YEARS OLD	7 (7%)
**FAMILY HISTORY (FOR PELVIC AND GASTROINTESTINAL TUMOURS)**	NEGATIVE	86 (86%)
	POSITIVE	14 (14%)
**HISTOTYPE**	ENDOMETRIOID	84 (84%)
	sEROUS	8 (8%)
	MIXED	7 (7%)
	UNDIFFERENTIATED	1 (1%)
**MYOMETRIAL** **INVASION**	<50%	52 (52%)
	≥50%	48 (48%)
**TUMOUR GRADE**	G1	20 (20%)
	G2	42 (42%)
	G3	38 (38%)
**FIGO STAGE**	IA	41 (41%)
	IB	26 (26%)
	II	13 (13%)
	IIIA	3 (3%)
	IIIB	7 (7%)
	IIIC	8 (8%)
	IV	2 (2%)
**LYMPHADENECTOMY**	NONE	55 (55%)
	PELVIC	24 (24%)
	PELVIC + PARA-AORTIC	17 (17%)
	BULKY	4 (4%)
**LVSI ^†^**	PRESENT	36 (36%)
	ABSENT	64 (64%)
**ADJUVANT TREATMENT** **(N = 99, 99%)**	SURVEILLANCE	51
	BRACHYTHERAPY	6
	EBRT ^‡^	12
	EBRT+BRACHYTHERAPY	7
	CT ^§^ +EBRT (“SANDWICH”)	16
	CT	4
	CT + EBRT	3
**NEOADJUVANT** **TREATMENT** **(N = 1, 1%)**		1

^†^ Lymphovascular space invasion; ^‡^ external beam radiotherapy; ^§^ chemotherapy.

**Table 2 jcm-11-05678-t002:** p53 status and histopathological features.

	p53 wt (*n* = 78)	p53 abn (*n* = 22)	Total (*n* = 100)	*p*-Value
**Histology**				0.0001
**− endometrioid**	70 (89.7%)	14 (58.3%)	84 (84%)	
**− non-endometrioid**	8 (10.3%)	8 (41.7%)	16 (16%)	
**Tumor grade**				0.004
**− G1**	17 (22.4%)	3 (12.5%)	20 (20%)	
**− G2**	37 (48.7%)	5 (20.8%)	42 (42%)	
**− G3**	22 (28.9%)	16 (66.7%)	38 (38%)	
**FIGO stage**				0.023
**− IA**	34 (44.7%)	7 (29.2%)	41 (41%)	
**− IB**	22 (28.9%)	4 (16.7%)	26 (26%)	
**− II**	10 (13.2%)	3 (12.5%)	13 (13%)	
**− III–IV**	10 (13.2%)	10 (41.7%)	20 (20%)	

p53 wt = p53 wildtype, p53 abn = p53 abnormal.

**Table 3 jcm-11-05678-t003:** Estrogen receptor (ER) status and histopathological features.

	ER− (*n* = 16)	ER+ (*n* = 84)	Total (*n* = 100)	*p*-Value
**Histology**				0.003
**− endometrioid**	9 (56.2%)	75 (89.3%)	84 (84%)	
**− non-endometrioid**	7 (43.8%)	9 (10.7%)	16 (16%)	
**Grade**				0.011
**− G1**	0 (0%)	20 (23.8%)	20 (20%)	
**− G2**	5 (31.3%)	37 (44%)	42 (42%)	
**− G3**	11 (68.8%)	27 (32.2%)	38 (38%)	
**FIGO stage**				0.013
**− IA**	2 (12.5%)	39 (46.4%)	41 (41%)	
**− IB**	6 (37.5%)	20 (23.8%)	26 (26%)	
**− II**	1 (6.3%)	12 (14.3%)	13 (13%)	
**− III–IV**	7 (43.8%)	13 (15.5%)	20 (20%)	

**Table 4 jcm-11-05678-t004:** Progesterone receptor (PgR) status and histopathological features.

	PgR− (*n* = 16)	PgR+ (*n* = 84)	Total (*n* = 100)	*p*-Value
**Histology**				0.029
**− endometrioid**	10 (62.5%)	74 (88%)	84 (84%)	
**− non-endometrioid**	6 (37.5%)	10 (12%)	16 (16%)	
**Grading**				0.003
**− G1**	0 (0%)	20 (23.8%)	20 (20%)	
**− G2**	4 (25%)	38 (45.2%)	42 (42%)	
**− G3**	12 (75%)	26 (31%)	38 (38%)	
**FIGO stage**				0.077
**− IA**	3 (18.8%)	38 (45.2%)	41 (41%)	
**− IB**	6 (37.5%)	20 (23.8%)	26 (26%)	
**− II**	1 (6.3%)	12 (14.3%)	13 (13%)	
**− III–IV**	6 (37.5%)	14 (16.7%)	20 (20%)	

## Data Availability

Data are available for consultation in any moment.

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
