# Peer review of "Immunohistochemical Markers and TILs Evaluation for Endometrial Carcinoma"

_jcm, 2022, doi:10.3390/jcm11195678_

Round 1

Reviewer 1 Report

This study proposes an alternative molecular classification bases on immunohistochemistry and TILS

I do not really understand why TILS are included in this study as few datas are available on their evaluation in contrast with demonstrated immunochemistry analyses of p53 etc…

Please include “retrospective” in your abstract

Why carcinosarcomas were excluded? Why did you not exclude clear cell or serous in this case? All European and non European recommendation do not have specific recommendation or classification for carcinosaromas

Your cohort represents few cases of endometrial cancers for a long period. How many endometrial cancers had surgery in your center during the period. I suppose many more, why only 100 patients were analyzed?

Methods used for TILS is really questionable

Using OS and PFS as primary endpoints is questionable.

Please identify which one is primary and secondary outcome. Usually, PFS as primary.

Mucinous histology is endometrioid cancer and should not be separated from endometrioid histology.

No sentinel lymph node procedures were performed during the study period? Were there included in pelvic lymphadenectomy?

Neoadjuvant treatment is not an adjuvant treatment. Please separate in table 1

What are you referring to for LVSI + and -? 1 LVSI is considered +?

Please add surgical routes 

Why did you include Figo stage in table 3 and note table 4 ?

The title mention “cost-effective”. In your article there is not any detail about costs. Please revise.

Comparisons between calcifications have to be tempered. No statistical associations are made in your study, and not possible as it is not the same population.

Reviewer 2 Report

1. All abbreviations used in the 1st time have to explained (full name) in abstract and main text. For example, in the abstract I cannot see the full name of MMR, OR etc.

2.Key words: not to use abbreviations

3. tables, figures are not self-explanatory.

4. the method and materials section was described insufficient. It has to be described desiring all steps of each method, provide name, city, country, catalog no. of each reagent used in this study.

5. 2.1. section has to be described the ethic.

6. please provide the calculated the size number using the statistical method. Does it sufficient in this study?

7. 3.1. section ought to be explain in the text in more details. It cannot be presented only  in the table.

8. Please add imagine of IHC staining results, including negative control. Describe it.

9. please provide expression of P53 by IHC

10. One method is a significant risk of limited study. Data needs to be analyzed in different methods too, i.e. WB. I am expecting to see this results.

11. Number of refs are not sufficient and need to be updated.

12. discussion is about nothing. It has to be improved, written in more scientific way.

13. the paper was not prepared according to the requirements.

14. CKNOWLEDGEMENTS, contribution-???????

Reviewer 3 Report

Dear Authors, 

I read with great interest the Manuscript titled "Immunohistochemical markers and TILs evaluation for endometrial carcinoma: a new cost-effective prognostic classification?". 

In my honest opinion, the topic is interesting enough to attract the readers’ attention. It is well written and the statistical method is correct. Nevertheless, you should make some minor changes. 

·       In the paragraph 2.2.1 you classified patients as high TILs level using as threshold the median value obtained from your results. It is better to refer to the recommendations provided by the International TILs Working Group 2014 for breast cancer (PMID: 25214542, PMID: 34656305).

·       In the paragraph 2.2.2, you considered MLH1 and PMS2 and MSH2 and MSH6 as partners. Please cite that in the literature it is described that the immunohistochemical expression of MMR can be assessed by considering the expression of MLH1, MSH2, MSH6 and PMS2, or only MSH2 and PMS2, since a combination of only two MMR may be used as a still cheaper test without affecting the diagnostic accuracy in detecting microsatellite instability (PMID: 33812697).

·       The discussion have to be improved: you have to start from the results of your paper, then you have to shown strengths and limitations of your work, comparison with the literature, clinical impact of your work and future perspectives.
